# Novel Strategies for Treating Castration-Resistant Prostate Cancer

**DOI:** 10.3390/biomedicines9040339

**Published:** 2021-03-27

**Authors:** David Ka-Wai Leung, Peter Ka-Fung Chiu, Chi-Fai Ng, Jeremy Yuen-Chun Teoh

**Affiliations:** S. H. Urology Centre, Department of Surgery, The Chinese University of Hong Kong, Hong Kong, China; davidleung1q2s3c@yahoo.com.hk (D.K.-W.L.); peterchiu@surgery.cuhk.edu.hk (P.K.-F.C.); ngcf@surgery.cuhk.edu.hk (C.-F.N.)

**Keywords:** prostate cancer, castration resistance, androgen deprivation therapy

## Abstract

The development of castration resistance is an inevitable pathway for the vast majority of patients with advanced prostate cancer. Recently, there have been significant breakthroughs in the understanding and management options of castration-resistant prostate cancer. Three novel hormonal agents showed survival benefits in non-metastatic patients. As for metastatic disease, there was an even wider range of management options being investigated. This review summarized advances in the management of castration-resistant prostate cancer (CRPC) including emerging data on novel imaging techniques and treatment strategies.

## 1. Introduction

Prostate cancer (PCa) is the second most commonly diagnosed cancer and the sixth leading cause of cancer mortality among men worldwide [1]. Despite the initial success of the androgen deprivation therapy (ADT) for advanced PCa, virtually all patients eventually develop biochemical and clinical evidence of treatment resistance. This disease status is known as castration-resistant prostate cancer (CRPC).

Research in genetics and cellular and molecular biology led to a better understanding of the mechanisms of castration resistance. Molecularly altered androgen receptors (AR) in CRPC could undergo activation by estrogens, progesterones, growth factors, and cytokines, even in the absence of androgens [2,3]. Recently, the overexpression of AR splice variants (in particular ARV7) was found to associate with CRPC and worse prognosis [4]. During the past decade, clinical investigators not only improved on diagnostic technology for metastatic diseases, but also established several categories of CRPC treatments: chemotherapy, novel hormonal agents, immuno-and-targeted therapy, and theranostics.

## 2. Definitions of CRPC and Novel Imaging

CRPC is defined as castrate serum testosterone levels (<50 ng/dL or 1.7 nmol/L) plus either biochemical or radiological progression, as specified in the European Association of Urology guidelines [5].

CRPC can take place in both non-metastatic and metastatic settings. The differentiation between non-metastatic CRPC (M0CRPC) and metastatic disease (mCRPC) is by conventional imaging, i.e., computed tomography (CT) and bone scan [5]. Furthermore, a prostate specific antigen-doubling time (PSA-DT) of less than 10 months is associated with a higher risk of bone metastases or death [6]. The median survival in mCRPC is approximately 35 months, depending on different prognostic factors and the use of second- and third-line systematic treatment [7].

Moreover, positron emission tomography of ^68^Ga-labelled prostate-specific membrane antigens (PSMA-PET) is a promising imaging modality in advanced prostate cancer. A systematic review studied the diagnostic accuracy of PSMA-PET performed in 1309 patients with advanced prostate cancer [8]. The sensitivity and specificity were both 86% on a per-patient basis. With the increased use of PSMA-PET, the accuracy for diagnosing early metastasis is expected to improve in the CRPC population.

For oligometastasis in hormone-sensitive and recurrent prostate cancer (typically defined as three or fewer metastases), the potential benefits of the primary tumor treatment and/or metastasis-directed therapy were explored [9]. Similarly, several retrospective studies suggested that ablative radiotherapy or surgery of oligometastases in CRPC might delay the PSA progression and the initiation of the next-line systematic treatment [10,11]. In this sense, oligometastatic CRPC appeared to be a distinct entity that warrants further investigations on its prognostic significance and implications on treatment strategies.

## 3. Treatment Options for M0CRPC

Three large randomized controlled trials (RCT), SPARTAN [12], PROSPER [13], and ARAMIS [14], evaluated the metastasis-free survival (MFS) as the primary end-point in patients with non-metastatic CRPC (M0CRPC) treated with enzalutamide (PROSPER), apalutamide (SPARTAN), and darolutamide (ARAMIS) against placebo, respectively. CT and bone scans were used in these trials to diagnose the non-metastatic status of the disease. Of note, only patients with a short PSA doubling time of fewer than 10 months were included. ADT was continuously used in both the novel agent arms and the placebo arms. MFS was defined as the time to the first metastasis on imaging or death.

Enzalutamide bound to AR with higher affinity than androgens, thereby inhibiting downstream nuclear translocation and DNA binding [13]. Apalutamide was another competitive AR inhibitor that reduced AR-mediated cancer growth [12]. Darolutamide shared similar mechanisms and had special chemical characteristics which prevented the drug from entering the blood-brain barrier [14].

In all of these trials, a significant MFS benefit was observed. Apalutamide yielded a median MFS of 40.5 months, compared to 16.2 months in the placebo group (homologous recombination (HR) 0.28; 95% CI: 0.23–0.35; *p* < 0.001) [12]. Enzalutamide gave a median MFS of 36.6 months vs. 14.7 months in the placebo arm (HR 0.29; 95% CI: 0.24–0.35; *p* < 0.001) [13]. Darolutamide was shown to have a median MFS of 40.4 months, compared to 18.4 months in the placebo group (HR 0.41; 95% CI: 0.34–0.50; *p* < 0.0001) [14].

The updated results of these trials presented in the 2020 American Society of Oncology (ASCO) meeting showed a significant overall survival (OS) benefit. In PROSPER, the median OS for the enzalutamide group was 67 months, compared to 56 months in the placebo group (HR 0.73; 95% CI: 0.61–0.89; *p* = 0.001). The benefit of enzalutamide was generally consistent across prespecified subgroups, with the potential exception of a small group of patients receiving bone-sparing agents [15]. In SPARTAN, apalutamide gave a better median OS than the placebo (73.9 vs. 59.9 months), corresponding to a relative reduction of 21.6% in the risk of mortality (HR 0.78; *p* = 0.0161) [16]. As for ARAMIS, with a median follow-up of 29 months, the 3-year OS rates were 83% and 77% on the darolutamide and placebo arms, respectively (HR 0.69; 95% CI: 0.53–0.88; *p* = 0.003). Notably, the use of darolutamide also significantly postponed the time of symptomatic bone events and the use of chemotherapy [17] compared to the placebo.

According to the above trials, these oral agents were generally well tolerated, with a treatment cessation due to adverse events in 9% for enzalutamide and darolutamide and 13.6% for apalutamide. Common side effects include fatigue (33% for enzalutamide, 31.9% for apalutamide, 16% for darolutamide), fall (11% for enzalutamide, 20.9% for apalutamide, 4% for darolutamide) and rashes (Not reached for enzalutamide, 24% for apalutamide, 3% for darolutamide). The clinical benefits and safety profiles of these drugs are summarized in Table 1.

## 4. Treatment Options for mCRPC

In the past, older-generation antiandrogens such as bicalutamide were the standard approach to treating mCRPC. Recent phase 3 studies, however, demonstrated better clinical outcomes for the use of chemotherapy (docetaxel and cabazitaxel), novel hormonal agents (abiraterone acetate and enzalutamide), Sipuleucel-T, Radium-223, and olaparib. Their key eligibility criteria and survival benefits are summarized in Table 2. Other investigational treatment modalities will also be discussed.

### 4.1. Bicalutamide

Before novel treatment regimens came into place, the addition of bicalutamide was commonly used to treat CRPC. The STRIVE trial compared enzalutamide with bicalutamide (a non-steroidal antiandrogen) in treating a total of 396 patients with M0CRPC and mCRPC. When compared with bicalutamide, enzalutamide significantly reduced the risk of mortality or disease progression by 76% (HR 0.24; 95% CI: 0.18–0.32; *p* < 0.001) and delayed the time of PSA progression [27]. TERRAIN is another randomized trial where enzalutamide was compared with bicalutamide in treating 375 mCRPC patients. The enzalutamide group had a significantly better median progression-free survival (PFS) than the bicalutamide group (15.7 months vs. 5.8 months; HR 0.44; 95% CI: 0.34–0.57; *p* < 0.0001) [28]. Therefore, the novel hormonal agents have better survival benefits for mCRPC patients.

### 4.2. Docetaxel

Chemotherapy was the first agent for mCRPC that gave survival benefits. Although mitoxantrone (a topoisomerase inhibitor) plus prednisolone was shown to relieve pain and improve health-related quality of life in mCRPC [29], it did not give any survival benefits [30]. The TAX 327 study was an RCT comparing docetaxel plus prednisolone with mitoxantrone plus prednisolone in 1006 mCRPC patients. As compared with the mitoxantrone group, the group given three-weekly docetaxel had a hazard ratio for death of 0.76 (95% CI: 0.62–0.94; *p* = 0.009), and the weekly docetaxel group had a hazard ratio for death of 0.91 (95% CI: 0.75–1.11; *p* = 0.36). The survival benefit of the three-weekly docetaxel was consistent across different subgroups defined by pain at the baseline, performance status, and age [18]. Serious adverse events occurred among 26–29% in the docetaxel group. Grade 3 or above neutropenia happened in 5% of patients who received docetaxel. Retrospective series showed that in mCRPC relapse after an initial good response to docetaxel, a rechallenge with docetaxel was more likely to yield a satisfactory PSA response and symptom relief than non-taxane therapy (40.4% vs. 10.6%; *p* < 0.001 for PSA), albeit no survival benefit [31].

Docetaxel is an anti-mitotic chemotherapy that binds to the beta-subunits in microtubules and causes apoptosis. Tumor cells in CRPC, however, often harbored multidrug resistance proteins, including P-glycoprotein (P-gp), multidrug resistance protein 1 (MRP1), and breast cancer resistance protein (BRCP) which disable drug binding and activate drug efflux. Moreover, genetic alterations in the apoptosis pathways, specifically upregulation of p53 and activation of PAR1 have been shown to reduce docetaxel-induced apoptosis [32].

### 4.3. Sipuleucel-T

Sipuleucel-T is a type of therapeutic oncological vaccine in which autologous blood mononuclear cells extracted from blood are activated ex vivo with a recombinant fusion protein (PA2024) before infusion, such that T-cell mediated cytotoxicity would preferentially attack prostate cancer cells harboring prostatic acid phosphatase. The IMPACT study included 512 men with mCRPC who had no visceral metastasis and not received prior systemic therapy other than ADT to receive sipuleucel-T or a placebo in a randomized manner. Upon a median follow-up period of 34 months, sipuleucel-T significantly reduced the mortality risk with a hazard ratio of 0.78 (95% CI: 0.61 to 0.98, *p* = 0.03). It also prolonged the median survival by 4.1 months (25.8 months vs. 21.7 months). Adverse events happened in 98.8% of the sipuleucel-T group, common ones being chills, pyrexia, back pain, and arthralgia. However, grade 3–5 complications occurred only in 31.7%, compared to 35.1% in the placebo group [19].

### 4.4. Abiraterone

Abiraterone acetate, which acts by inhibiting androgen production, is proven to have survival benefits for mCRPC with prior use of docetaxel. COU-AA-301 is an RCT in which 1195 mCRPC patients with progression after docetaxel and no visceral metastasis were randomized to receive either abiraterone plus prednisolone or placebo plus prednisolone. At the median follow-up of 20 months, the abiraterone group outperformed the placebo group in terms of median OS (15.8 months vs. 11.2 months; HR 0.74; 95%; *p* < 0.0001), median time to PSA progression (8.5 months vs. 6.6 months; HR 0.63; 95% CI: 0.52–0.78; *p* < 0.0001), and median radiologic PFS (5.6 months vs. 3.6 months; HR 0.66; 0.58–0.76; *p* < 0.0001). Common grade 3–4 adverse events included fatigue (9% for abiraterone, 10% for placebo), back pain (7% vs. 10%), anemia (8% in both groups), and bone pain (6% vs. 8%) [21]. COU-AA-302 included 1088 mCRPC patients who had not received chemotherapy and randomized them to receive either abiraterone plus prednisolone or prednisolone only. Over a median follow-up period of 22 months, abiraterone improved radiologic PFS (16.5 months vs. 8.3 months; HR 0.53; 95% CI: 0.45–0.62; *p* < 0.001), and gave a trend towards OS benefit (NR vs. 27.2 months; HR 0.75; 95% CI: 0.61–0.93; *p* = 0.01), though not reaching the prespecified boundary of significance (*p* < 0.001). As for secondary endpoints: abiraterone significantly delayed the time to PSA progression (11.1 months vs. 5.6 months; HR 0.49 (0.42–0.57); *p* < 0.001) and time to chemotherapy (25.2 months vs. 16.8 months; HR 0.58 (0.49–0.69); *p* < 0.001) [20].

### 4.5. Enzalutamide

Enzalutamide is a promising AR targeting agent in the management of mCRPC in both chemotherapy-naïve and post-chemotherapy settings. The AFFIRM study randomized 1199 men whose CRPC progressed after chemotherapy. The median overall survival was 18.4 months in the enzalutamide group versus 13.6 months in the placebo group, representing an HR of 0.63 (95% CI: 0.53–0.75; *p* < 0.001). Enzalutamide was significantly superior to the placebo with regards to all secondary endpoints: the rate of ≥50% drop in PSA (54% vs. 2%; *p* < 0.001), the time to the first bone event, quality-of-life, the time to PSA progression, and time to radiographic PFS. As for side effects, the rates of fatigue, diarrhea, and hot flashes were higher in the enzalutamide group. Seizures were reported in five patients (0.6%) receiving enzalutamide [23]. The PREVAIL study randomized 1717 mCRPC patients with no prior chemotherapy to receive either enzalutamide or placebo. The benefit of enzalutamide was found to persist in OS, radiographic PFS, and all secondary endpoints. At median follow-up of 22 months, enzalutamide improved median OS (32.4 months vs. 30.2 months; HR 0.71; 95% CI: 0.60; *p* < 0.001) [22]. An updated analysis of PREVAIL data published in 2017 showed that enzalutamide reduced the risk of death by 23% compared to the placebo (HR 0.77; 95% CI: 0.67–0.88; *p* = 0.0002). At the median follow-up of 31 months, the median OS was 35.3 months (95% CI: 32.2—not yet reached) in the enzalutamide arm and 31.3 months (95% CI: 28.8–34.2) in the cohort originally randomized to placebo. In addition, enzalutamide reduced the risk of radiological progression or mortality by 68% compared to the placebo (HR 0.32; 95% CI: 0.28–0.37; *p* < 0.0001) [33]. According to a review on sequencing of mCRPC treatment, crossover from abiraterone to enzalutamide and vice versa, gave a radiographic response in merely 0–8% of all patients [34]. Therefore, due consideration should be given to other non-AR-pathway treatment options if one of these drugs already failed.

As of now, there have not been published phase 3 data supporting the use of apalutamide and darolutamide in mCRPC. However, there is an active ongoing research in this field. For instance, ODENZA is an ongoing phase 2 study comparing enzalutamide with darolutamide in mCRPC.

The possible reasons for treatment failure of abiraterone and enzalutamide were investigated. Androgen receptor (AR) splice variants represented a diverse group of molecules that can replace the functions of androgen receptors. A prospective trial demonstrated that 20–40% of circulating tumor cells in CRPC patients treated with abiraterone and enzalutamide had an active AR splice variant 7 (AR-V7), which was significantly associated with a lower PSA response rate and shorter PFS and OS compared to men without AR-V7 expression [35]. This common mechanism might explain why switching between these two drugs typically gives a poor response. In addition, the use of these novel hormonal agents also led to an increase in AR-independent prostate cancers. Recent molecular studies indicated an overexpression of neuroendocrine markers such as transcription factor-4 in enzalutamide-refractory prostate cancers [36]. Therefore, neuroendocrine (NE) differentiation of prostate cancer may be attributable to treatment resistance of novel hormonal agents. Bluemn et al. showed that the incidence of double-negative (AR-negative and NE-negative) prostate cancer had risen in the past two decades. This subset of the disease tends to harbor hyperactivity in other molecular (FGF, MAPK) pathways [37]. Better understanding of these AR-independent pathways may lead to the development of molecular-modulating agents in the future.

### 4.6. Cabazitaxel

Cabazitaxel is a taxane drug with antitumor activity in treating mCRPC. The TROPIC trial compared cabazitaxel with mitoxantrone in treating 755 patients whose mCRPC had progression despite undergoing docetaxel therapy. Cabazitaxel demonstrated significant improvements in the median OS (15.1 months vs. 12.7 months; HR 0.7) and PFS (2.8 months vs. 1.4 months; HR 0.74). The most common clinically significant grade 3 or higher adverse events were neutropenia (82% in cabazitaxel vs. 58% in mitoxantrone) and diarrhea (6% in cabazitaxel vs. <1% in mitoxantrone) [24]. However, no survival benefit over docetaxel was shown when cabazitaxel was used in chemotherapy-naïve mCRPC [38]. The CARD trial compared cabazitaxel with the other AR-targeting agents (abiraterone or enzalutamide) in treating 255 patients whose mCRPC had progressed after docetaxel and one of the inhibitors. Compared to AR-targeting agents, cabazitaxel gave better OS (13.6 months vs. 11 months; HR 0.64; *p* = 0.008), better median PFS (4.4 months vs. 2.7 months) and higher PSA response rates (35.7% vs. 13.5%). The hazard ratio for progression or mortality was 0.52 (95% CI: 0.40–0.68; *p* < 0.001) [39].

### 4.7. Radium-223

Theranostics refers to companion agents targeting a specific biological entity, usually tumor cells, for diagnostic and/or therapeutic purposes. Radium-223 is an alpha emitter that preferentially attaches to areas with increased bone turnover, delivers alpha-particle radiation to bone metastases, and it has a role in the modern management of CRPC with bone metastases. The ALSYMPCA study randomized over 900 mCRPC patients with two or more bone metastases, and no known visceral metastasis, receiving either radium-223 or the standard of care. Radium-223 significantly improved median OS (14.9 months vs. 11.3 months; HR 0.70; 95% CI: 0.58–0.83; *p* < 0.001). A benefit of radium-223, compared to placebo, was also proven in secondary endpoints: the time to the first symptomatic skeletal event (15.6 months vs. 9.8 months; HR 0.66; 95% CI: 0.52–0.83; *p* < 0.001), the time to an increase in the total alkaline phosphatase level (HR 0.17; *p* < 0.001), and the time to PSA rise (HR 0.64; *p* < 0.001). For adverse effects, radium-223 was not associated with more grade 3 or above complications when compared to placebo [25]. A prespecified subgroup analysis from ALSYMPCA demonstrated that the benefit of radium-223 persisted, irrespective of previous chemotherapy use [40].

### 4.8. Lutetium-177

Lutetium-177 (177Lu)-PSMA-617 represents a type of radiolabeled small ligand molecules that bind with high affinity to the prostate-specific membrane antigen (PSMA), thereby enabling beta particle therapy targeted to mCRPC. In the phase 2 single-arm LuPSMA study, 30 patients (87% of whom received at least one line of therapy including chemotherapy, abiraterone, enzalutamide, or both) received this novel treatment. An objective response (OR) was achieved in 14 out of 17 patients (82%) with measurable disease in lymph nodes or viscera. There was also a significant improvement in pain scores across all timepoints. Moreover, eleven patients (37%) experienced a ten-point or more improvement in the global health score [41]. In this regard, there are several ongoing trials including ANZUP (phase 2 study comparing (177Lu)-PSMA-617 with cabazitaxel) and VISION (phase 3 trial of (177Lu)-PSMA-617 vs. best supportive care), whose results are eagerly awaited.

### 4.9. PARP Inhibitors

Poly (ADP-ribose) polymerase (PARP) is an enzyme that regulates DNA repair pathways. PARP inhibitors can cause synthetic lethality in tumor cells with pre-existing homologous recombinant repair (HRR) gene mutations such as *BRCA2*, *BRCA1*, and *ATM*. Recently, two PARP inhibitors, namely olaparib and rucaparib, have come into play for use in HRR gene-mutated mCRPC. The PROfound study evaluated the efficacy of olaparib against that of the physician’s choice of either abiraterone or enzalutamide in mCRPC patients who had disease progression while receiving a new hormonal agent (enzalutamide or abiraterone). Among those with at least one alteration in BRCA1, BRCA2, or ATM, olaparib resulted in a better radiological PFS, with an HR for disease progression or mortality of 0.34 (95% CI: 0.25–0.47; *p* < 0.001). There was, however, no significant improvement in the overall survival [26]. On the other hand, there was no significant benefit in PFS for the combined cohort of patients harboring any of the 15 prespecified HRR gene mutations in the PROfound study [42]. TRITON2 was another phase 2 single-arm study investigating the use of rucaparib in mCRPC with disease progression despite one novel AR-targeting agent and taxane-based chemotherapy. Among the evaluable patients with a *BRCA1* or *BRCA2* alteration, the objective response rate by independent radiologists was 43.5% (95% CI: 31.0–56.7%; 27 of 62 patients), and the confirmed PSA response rate was 54.8% (95% CI: 45.2–64.1%; 63 of 115 patients) [43]. Common adverse effects of PARP inhibitors included anemia, thrombocytopenia, and neutropenia [44]. The TOPARD-B trial selected mCRPC patients who had received taxane-base chemotherapy with genetic sequencing of prostate cancer biopsies showing DNA damage response (DDR) gene aberrations, and prospectively randomized them to 300 mg or 400 mg olaparib twice daily. The primary endpoint was achievement of any of the following outcomes: a radiological objective response, a ≥ 50% drop in PSA, or a conversion of the circulating tumor cell count to <5 cells per 7.5 mL blood. In the 400 mg cohort, 25 out of 46 (54%) achieved this composite response; in the 300 mg cohort, 18 out of 46 patients (39%) achieved the response [45].

The use of lutetium-177 and rucaparib in the management of mCRPC has been summarized in Table 3.

### 4.10. Pembrolizumab

Pembrolizumab has shown anti-tumor activity in programmed death-ligand-1 (PD-L1)-positive mCRPC. The phase 2 KEYNOTE-199 study included 258 mCRPC patients treated with docetaxel and novel AR-targeting agents. Cohorts 1, 2, and 3 enrolled patients with radiologically evaluable PD-L1-positive, PD-L1-negative disease, and bone-predominant disease (regardless of PD-L1), respectively. Among the 133 patients in cohort 1 and 66 in cohort 2, the objective response rate (ORR) was 5% (95% CI: 2–11%) and 3% (95% CI: <1–11%), respectively. The median OS for cohorts 1, 2, and 3 were 9.5 months, 7.9 months, and 14.1 months, respectively. Adverse effects happened in 60% of patients. Grade 3 to 5 adverse events happened in 15%, and 5% discontinued treatment because of side effects [46]. Furthermore, combination therapy for mCRPC is under investigation. The KEYNOTE-641, for instance, is an ongoing randomized phase 3 trial comparing pembrolizumab plus enzalutamide with placebo plus enzalutamide in mCRPC.

### 4.11. Bone Protection in CRPC

Bone metastases (BM) are a common cause of morbidity in patients with prostate cancer. If left untreated, BM can lead to disabling pain, pathological fracture, and spinal cord compression. In this regard, the use of bisphosphonates and Receptor activator of nuclear factor kappa-Β ligand (RANKL) inhibitors was investigated to prevent skeletal-related events (SRE) in advanced prostate cancer. The Zometa 039 trial found that a small proportion of patients receiving zoledronic acid had SRE, compared with the placebo group (33.2% vs. 44.2%; difference = 11.0%; *p* = 0.021). The median time to first SRE was delayed with the use of zoledronic acid 4 mg (321 days vs. NR, *p* = 0.011). There was a significant improvement in pain scores but not in disease progression or quality-of-life scores [47]. In an RCT where 1904 CRPC patients were randomized into receiving zoledronic acid or denosumab (an anti-RANK-ligand antibody), the median time to SRE was 20.7 months with denosumab, compared with 17.1 months with zoledronic acid (HR 0.82; *p* = 0.0002). Serious adverse events happened in 63% of the denosumab group and 60% of the zoledronic acid group. Hypocalcemia took place more frequently in the denosumab group (13%) than in the zoledronic acid group (6%). Osteonecrosis of the jaw occurred infrequently in 1–2% of the patients [48]. Therefore, the authors commented that denosumab was superior to zoledronic acid for prevention of SRE in CRPC.

## 5. Recommendations for Genetic Testing in Advanced Prostate Cancer

The updated 2019 National Comprehensive Cancer Network (NCCN) Guidelines [49] recommend considering tumor testing for homologous recombination (HR) mutations and microsatellite instability (MSI) or deficient DNA mismatch repair (dMMR) gene mutations among patients with either regional spread or distantly metastasized prostate cancer. Germline mutation testing should be recommended in all men diagnosed with NCCN high-risk or metastatic prostate cancer. Such testing can inform treatment decision-making. The United States Food and Drug Administration (FDA) issued accelerated approval in 2017 for pembrolizumab-programmed cell death-1 (PD-1) immune checkpoint inhibition for patients with unresectable or metastatic MSI-high or dMMR cancers. It was the first approval for a cancer treatment based not on cancer type but mutations. The presence of HR mutations opens the possibility of PARP inhibitors in the second-line treatment for mCRPC. According to the 2019 Philadelphia Prostate Cancer Consensus Conference, metastatic disease or family history suggestive of hereditary PCA was recommended for germline testing. Priority genes to test for metastatic disease treatment included BRCA1, BRCA2, and dMMR genes, whereas broader testing such as ATM is reserved for clinical trial eligibility [50].

## 6. Conclusions

CRPC can be considered the final common pathway for all advanced prostate cancers. With the advancement in imaging technology, the diagnostic accuracy for M staging in CRPC markedly improved. While several novel AR signaling agents showed promising benefits in M0CRPC, the wide spectrum of treatment options for mCRPC can be classified into novel hormonal agents, chemotherapy, immunotherapy, and theranostics. Moreover, a better understanding of the molecular biology of PCa contributed to the advent of more targeted and gene-specific therapies. Therefore, genetic testing of prostate tumors may offer valuable perspectives in choosing the most appropriate treatment.

## Figures and Tables

**Table 1 biomedicines-09-00339-t001:** Summary of non-metastatic castration-resistant prostate cancer (M0CRPC) treatment options.

Study	SPARTAN [12]	PROSPER [13]	ARAMIS [14]
Agent	Apalutamide	Enzalutamide	Darolutamide
Dosage	240 mg daily	160 mg daily	600 mg BD with food
MFS (months)	40.5 vs. 16.2 HR 0.28; *p* < 0.0001	36.6 vs. 14.7 HR 0.29; *p* < 0.001	40.4 vs. 18.4 HR 0.41; *p* < 0.0001
Updated OS (months)	73.9 vs. 59.9 HR 0.78; *p* = 0.016	67 vs. 56.3 HR 0.73; *p* = 0.001	83 vs. 77 HR 0.69; *p* = 0.003
Adverse event (AE) reporting	Every 1 month	Every 4 months	Every 4 months
Grade ¾ AE (%)	53	31	25
Fatigue (%)	31.9	33	16
Fall (%)	20.9	11	4
Rash (%)	24	NR	3
Treatment cessation due to AE (%)	13.6	9	9

OS, significant overall survival; HR, homologous recombination; NR, not reached; MFS, metastasis-free survival.

**Table 2 biomedicines-09-00339-t002:** Summary of established metastatic castration-resistant prostate cancer (mCRPC) treatment options.

Study	Agent	Control	Sample Size	Indication	HR	OS Benefit (months)
TAX-327 [18]	Docetaxel + Prednisolone	Mitoxantrone + Prednisolone	1006	mCRPC, symptomatic or not	0.76	2.9
IMPACT [19]	Sipuleucel-T	Placebo	512	mCRPC (pre-chemotherapy) mild/no symptoms, no visceral metastasis	0.78	4.1
COU-AA-302 [20]	Abiraterone + Prednisolone	Prednisolone	1088	mCRPC (pre-chemotherapy) mild/no symptoms, no visceral metastasis	0.81	NR
COU-AA-301 [21]	Abiraterone + Prednisolone	Prednisolone	1195	mCRPC (post-chemotherapy)	0.74	4.6
PREVAIL [22]	Enzalutamide	Placebo	1717	mCRPC (pre-chemotherapy)	0.77	4.0
AFFIRM [23]	Enzalutamide	Placebo	1199	mCRPC (post-chemotherapy)	0.63	4.8
TROPIC [24]	Cabazitaxel + Prednisolone	Mitoxantrone + Prednisolone	755	mCRPC (post-chemotherapy)	0.70	2.4
ALSYMPCA [25]	Radium-223	Placebo	921	mCRPC (post- or unfit for chemotherapy)	0.70	3.6
PROFOUND [26]	Olaparib	Enzalutamide or Abiraterone	387	mCRPC disease progression after either enzalutamide or abiraterone	0.34	N/A

**Table 3 biomedicines-09-00339-t003:** Summary of the other mCRPC treatment options.

Study	Agent	Sample Size	Indication	Benefit
LuPSMA [41]	Lutetium-177	30	mCRPC having received either one line of therapy	82% ORR in nodal and visceral disease; improved pain scores
TRITON2 [44]	Rucaparib	190	mCRPC with progression despite one novel AR-targeting agent and chemotherapy	43.5% ORR; 54.8% PSA response in the BRCA1/2 alteration subgroup

ORR, objective response rate.

## Data Availability

Not applicable.

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
