# Peer review of "Novel Strategies for Treating Castration-Resistant Prostate Cancer"

_biomedicines, 2021, doi:10.3390/biomedicines9040339_

Round 1
Reviewer 1 Report
In the review article entitled “Novel Strategy for Treating Castration-Resistant Prostate Cancer”, David Ka-Wai Leung et al summarized the management options of castration-resistant prostate cancer. The manuscript is well written. The authors present an unbiased summary and a critical discussion on the established and developing treatment modalities of castrate-resistant prostate cancer.
Author Response
Your comments are well noted. Thank you.
Reviewer 2 Report
In this review, Leung et al. provide a summary and review of current treatment options for advanced prostate cancers. The authors present information on new imaging technologies, new treatment options for M0CRPC, treatment options for mCRPC and advances in genetic testing. The review article, for the most part, is written appropriately and presented in a manner that is easy to read, and would be of interest to readers.
Major comments
- At the beginning of “3. Treatment options for M0CRPC”, it would be helpful to briefly include the mechanisms of action of the 3 drugs described.
- On page 3, line 95, the statement “In the past, antiandrogens were the standard approach to treating mCRPC” is misleading, as second-generation androgen deprivation treatments such as enzalutamide are also anti-androgens. Line 102 similarly suggests that anti-androgens are no longer used.
- The last paragraph on p.5 should include information on how these new treatments have also led to an increase in AR independent prostate cancers such as neuroendocrine and double-negative prostate cancers.
- Since the use of Apalutamide and Darolutamide are described for M0CRPC, it would be more consistent to include information on these drugs on p. 11 for mCRPC.
Minor comments
- The authors should consider revising the title to “Novel Strategies …” as more than one strategy is presented.
- The tables are very helpful. It would be helpful to add the citation reference numbers to the tables to direct readers to the studies.
- The table number in line 99 should be corrected to Table 2.
- The authors should consider changing the title “3.1. Older Generation Anti-Androgens” to Bicalutamide.
- In Table 2, it is not clear how the OS benefit for the COUT-AA-301 study is 4.4 months, if the end-point was not reached (line 167).
- In Table 2, the OS benefit for the ALSYMPCA study is listed as 2.8 months, whereas 14.9 months vs 11.3 months = 3.6 months (line 237).
- A revision of lines 239-240 would be helpful.
Author Response
Thank you for your comments. Changes have been made accordingly.
Major comments
- The mechanisms of action of the 3 drugs were added.
- Bicalutamide was used to replace antiandrogens.
- Discussion on neuroendocrine and double-negative prostate cancers has been added.
- The current status for Apalutamide and Darolutamide in mCRPC was described.
Minor comments
- “Novel Strategies" was used In the title.
- Reference numbers have been added to the tables.
- The table number was corrected to Table 2.
- The subtitle “3.1. Older Generation Anti-Androgens” was changed to Bicalutamide.
- In Table 2, the OS benefit for the COUT-AA-301 study was clarified to NR.
- In Table 2, the OS benefit for the ALSYMPCA study was amended to 3.6 months.
- The duplication of words was amended.
Reviewer 3 Report
The manuscript “Novel Strategy for Treating Castration-Resistant Prostate Cancer” by Dr DKW Leung et al presents a summary of current treatments opportunities of castration resistant prostate cancer. The text is covering, systematic ad concise, and such could serve as a quick and useful summary for browsing the current situation of the therapeutic approaches to treatment, and the recent and ongoing clinical studies evaluating the drugs and drug combinations. The text also contains brief comments on the mechanisms of action of the therapeutics. Although the focus is not on their description, they could be written in a bit more precise and up-to-date way. As an example I would like to mention a description of the mechanisms of AR in CRPC (Introduction, lines 2-4). The information is picked up from excellent summaries (Refs 2 and3) from behind two decades but the mechanisms have been specified since that, which could be taken into account in this review as well. Another example concerns a description of Lutetium-177 [177Lu]-PSMA-617, a radiolabelled small ligand (or inhibitor) molecule which binds with high affinity to prostate-specific membrane antigen (PSMA) etc.
Author Response
Thank you for your comments.
The mechanisms for AR in CRPC and also novel treatment e.g. Lu-177 have been updated accordingly.
Round 2
Reviewer 2 Report
All concerns have been addressed.